# Disinformation and Verification in a Digital Society: An Analysis of Strategies and Policies Applied in the European Regional TV Broadcasters of the CIRCOM Network

Jose Rúas-Araújo, Talia Rodríguez-Martelo * and Julia Fontenla-Pedreira

Faculty of Communication, Campus of Pontevedra, University of Vigo, 36005 Pontevedra, Spain
* Correspondence: talia.rodriguez@uvigo.es

**Abstract:** The recent COVID-19 health crisis has shone a spotlight on disinformation as the circulation of false information became more and more prominent. What the World Health Organization (WHO) has defined as an 'infodemic' poses a great risk for democracies and for society in general. In this context, public television channels, with their regional scope, actively participate in the fight against misinformation. This research aims to identify and classify the different verification initiatives and technological tools, as well as the different strategies and codes used in fact-checking tasks by European broadcasters belonging to the CIRCOM network. The methodology undertakes an exploratory approach and employs a questionnaire that is applied to a sample of the members of the network. Managers and professionals with executive profiles were asked about the management, operation and strategies used in the verification process. In light of the results obtained, it can be concluded that the current verification processes are based on human efforts, rather than technological tools, amounting to a total dependence on content curation by the writing teams in the newsroom. Thus, it is evidenced that in most cases, there is neither a specific department for verification, nor sufficient resources, despite the fact that all those surveyed regard disinformation as a priority issue, a threat to democratic integrity and a responsibility of public service media.

**Keywords:** disinformation; verification; CIRCOM; broadcasters; European; regional TV

## 1. Introduction and Literature Review

For several years, the media have been stepping up their fight against disinformation. Recent events in the political, health and social spheres, such as the assault on the Capitol in the United States [1], the global pandemic with the phenomenon of infodemia [2,3] or even the recent Russian–Ukrainian war [4] have demonstrated the great power and harmful effects of disinformation. The circulation of fake news through social networks and digital media has reached unprecedented heights. In this sense, the trust placed by citizens in institutions and the media is undermined [5]. For this reason, the verification of information is considered a core task of current newsrooms [6]. In this context, verification and the fight against misinformation are key in the development of public service media (PSM). These media are usually defined as corporations or broadcasters that are publicly owned, financed and controlled by the public. Their output is to inform, educate and entertain the audience [7]. The defense of diverse and multicultural media environments is closely related to the importance placed in veracity as a public value, and the right to the pluralism of information for citizens. For this reason, regional public media enjoy great relevance in regards to this matter as they aim to defend the culture of the communities in which they operate. They also seek to establish and maintain emotional bonds with their audiences. Moreover, these communities are linked to each other and to wider environments because the current facilities aim to be connected and enlarge its broadcasting [8].

The European Association of Regional Television (CIRCOM) is made up of 43 stations belonging to 29 European countries. These are publicly owned regional broadcasters that

are affiliated through this entity, with the aim of promoting and developing cooperation among their members, co-producing and exchanging content and strengthening vocational training for journalists and technicians across Europe [9].

## 2. Disinformation and Verification on Television

In the field of traditional media, specifically in radio and television, disinformation has seemingly severely discredited news programs in the eyes of the viewers [10]. According to data from the Reuters Institute's Digital News Report, less than 50% of respondents all around Europe trusted the news in the 2022 report, a lower score than the one from the previous year [11].

However, television audiences and trust in television increased during the months of confinement, in particular with regard to television news programs [12]. This occurred despite the changes in the production of content and formats derived from the limited resources resulting from the restrictions [13].

This boost given by the pandemic to radio and television has not been enough to rebuild the credibility and stability of the audiences of the past. The phenomenon of misinformation is accompanied by that of platformisation and the break with linear broadcasting [14].

The pressure exerted by the platforms on the foregoing audiovisual models leads, inevitably, to a significant change in the way of viewing, the dialogue with the audience and the format of the content [15,16]. The internationalization of media companies and the homogenization of content through similar and substitute proposals makes it very difficult for linear television to compete with the current offer [17]. Despite the global reach achieved in the past years, the direction is not clear since new and old audiences with very different habits, abilities and tastes have to coexist all over this context [18].

In this environment, PSM are forced to innovate and keep up to date in order to attract new audiences without losing already-established viewers. In addition, they must set up a dialogue with new forms of consumption and experiment with the multiple opportunities offered by interactivity for the participation of the spectators. At the same time, they continue to be ruled by their nature as a public service that requires them to guarantee values such as universality, diversity, creativity and innovation [19]. The regional perspective accentuates the engagement with the audience. This is because the connection with spectators and local entities is established by proximity and the development of community identity.

Today, convergence is part of the circulation of audiovisual programs and products in such a way that there is no single access route or means. This characteristic of information as something circular and non-linear determines a pattern of consumption in which misinformation finds easy and fast paths to spread [20].

In this sense, verification has become one of the newsrooms' fundamental tasks. The information that has to be contrasted systematically requires a great deal of effort. A significant investment of human and technological resources is needed to process the volume of information that is disseminated through social networks before reaching the traditional media [21].

Despite the fact that there are increasingly better and more numerous technological tools to manage verification processes, the reality of newsrooms is that the work of checking information, even if it has been something inherent to the journalistic event itself, requires a great investment in time and knowledge [22]. The sophistication of technological resources for the creation of disinformation requires the journalists to be up to date. This means keeping their training in new technologies upgraded in order to be able to handle the current volume of false information [23].

Collaboration with independent initiatives outside of the media system is now not only desirable but essential. In addition, these fact-checking platforms that operate as verifiers of the most widely distributed hoaxes are also accountability tools. Directly or indirectly, they monitor the false content that reaches radio and television stations [24].

PSM have a strong commitment to the distribution of truthful information. They also develop an arbitration of the circulation of false information which serves as a thermometer of the degree of disinformation that citizens are receiving and how it is replicated through social networks [25]. Given that, the public service media guarantees their independence by making the fight against fake news a strategic key that gives meaning to veracity as a public value. This is one of the most important issues that differentiates publicly owned and private media, which is their approach to and development of relevant ideas to improve society [26].

Disinformation is a global problem with interest at local and regional levels [27]. The way false information is filtered and its massive and individual distribution through social networks and digital platforms makes its traceability a remarkable question to research [28]. In this context of proximity given by regional media, it is possible to observe with greater accuracy how specific information affects the population and to what extent.

Television is accused of having fallen victim to false information on two fronts. Firstly, because the audiovisual medium is presently used more frequently to circulate fake news [29]. Secondly, because of the resonance of the information that constantly comes from the same sources, making it difficult to compare information. In recent years, the precariousness of the media has resulted in layoffs, scarcity of resources, the convergence of content and the homogenization of the media agenda [30].

## 3. European Regional Television Stations of CIRCOM Network

The CIRCOM network has been active for fifty years. In 1973, the beginning of what would eventually be the current association and from which it takes its name, the International Cooperative for Research and Action in Communication [1], was established. During the Prix Italia in Venice, a small group of professionals from public television met with Pierre Schaffer, a member of the Audiovisual High Council, Director of the Research Department of ORTF, France. Their initiative aimed to accelerate the development of regional identities. In 1983, the creation of CIRCOM Regional took place following the success of the debate organised by CIRCOM during the Prix Italia in Riva in 1980 on the theme "Regional Television: the last network", based on the growing awareness that regional television would play a key role in the development of the European idea. The founding members were Belgium, France, Germany and Italy. Registered since May 1995 as an Association at the Court of Strasbourg, benefiting from the local Code of Law, the Association is composed of the European Board Executive Committee with one National Coordinator from each member country, up to five individual members elected by the Board. These members have a main goal, which is to create a think tank for the exchange of ideas and collaborative work. Considering that goal, the network tries to examine regional television in Europe from an innovative and practical angle and contribute to the development of regional culture and identities, to bring together researchers and professionals from the mass media across borders and initiate a dynamic cultural approach in regional development, to provide a unique forum for ideas and experience and a network for exchanging personnel and equipment between European regional stations, to foster communication between members and make coproductions and encourage the exchange of regional programmes: theme programmes, news magazines, cross-border news bulletins, documentaries, programmes for young people and cultural and music programmes [9].

Given that the CIRCOM network is publicly owned and promotes regional television, the battle against disinformation is of key importance. This commitment has materialized through a long-running and intense training program for journalists.

Thus, in 2019, a workshop on "Fake News and news contrast" was held in Rome. This event was attended by media and journalists from Bulgaria, Germany, Italy, the Czech Republic, France, Malta, Serbia, Catalonia and Galicia [31]. In the year 2020, the seminar "Fact-checking at the Regional Scale" was held online, with a focus on the impact of verification at the regional level. Again in 2021, within the contents of the training

program "Learning from a global pandemic", disinformation was the central theme in the session "Fighting fake news" [32].

Public television stations offer their own verification mechanisms, spaces and programs, or those in collaboration with fact-checking initiatives already in operation in their respective geographical areas. This conveys to the audience the importance of verified information on the most rampant hoaxes coming from social media and from other media outlets [33].

Belgium's RTBF provides extensive content on disinformation from a media literacy approach. They have developed the Faky [34] tool, a website where citizens can check the reliability of information by entering the links on their platform. This initiative highlights the importance of collaboration between media, verifiers and viewers as a way to stop the circulation of false information. France, one of the countries with the most stations associated with the network, develops various contents focused on verification resulting from collaboration with different media. This includes the weekly program "Vrai ou Fake" based on information verified by assorted public broadcasters in France [35].

Participation in projects with other parties or in collaboration with other media outlets is very effective when it comes to combining resources to generate platforms. This can be seen in the Norwegian NRK and the Faktisk project in which VG, Dagbladet, NRK, TV2, Polaris media and Amedia participate.

Spain is the country in the CIRCOM network with the largest number of regional channels, amounting to a total of eleven stations: CRTVG (Galician Television), EITB (Basque Public Radio and Television), EPRTVIB (Radio and Television of the Balearic Islands), RTPA (Radio and Television of the Principality of Asturias), RTVCyL (Castilla Y León Television), CARTV (Aragonese Radio and Television Corporation), CCMA (Catalan Audiovisual Media Corporation), RTRM (Radiotelevision of the Region of Murcia), CEXMA (Extremadura Audiovisual Media Corporation), CMM (Castilla-La Mancha Media) and Telemadrid. The Galician station, CRTVG, developed the institutional campaign "*Coidémonos*" ("Let's take care of each other") to prevent misinformation associated with the pandemic. The Basque station EITB broadcasted a weekly section to warn of possible misinformation, in addition to the inFORMAZIOA training program regarding the detection of false information and the "Coronabulos" (Coronavirus hoaxes) initiative through which citizens were given the opportunity to send suspicious information for verification [36].

In most cases, these noteworthy examples are based on citizen collaboration as an essential element in the construction of the different platforms, solutions or television content [37]. It is part of the fundamental vision of the members of the network to connect with both their traditional and young audiences. Indeed, the interaction with their environments is key in understanding the role of regional public service media.

## 4. Methodology

The main objective of this research is to determine which verification strategies are currently applied by the public broadcasters of the selected sample made up of the Association of Regional Public Service Television in Europe, CIRCOM [2]. For this, a descriptive-analytic approach has been chosen, together with a qualitative approach, based on a questionnaire created as part of the 38th Annual Conference held in Galway (IR), on 26 and 27 May 2022.

This event is held annually. A series of talks and debates on current affairs in the audiovisual sector are performed, creating a place for the exchange of ideas. In every case, the conference is hosted by a different channel and city who acts as the organizer changed too. The topics covered in each edition are wide and diverse but are always focused on strategic issues such as content, technology, audience, online production and other broadcasting matters. The conference content is produced around a theme that is topical for the regional host and the industry. The sessions are produced by the members of CIRCOM Regional [38].

In this environment, the attendees are workers from CIRCOM member channels with managerial and executive profiles involved in the areas of management, innovation, programming, etc. It is also an interesting meeting point for other strategic stakeholders such as content creators, producers, distributors and other activities involved in the television business.

The interview program was established in coordination with the head of the organization. This allowed us to reach the managers of every attending broadcaster in order to carry out a questionnaire on disinformation, verification and digitization.

The questionnaire consisted of nine questions, five of an open and qualitative nature, and four of a dichotomous, closed response. This aims to interpret both the qualitative and quantitative results obtained through the responses of the participants interviewed.

A series of descriptive interpretations of the fight against disinformation, and of the verification strategies applied within the framework of the European regional media, was subsequently obtained. Content analysis allowed for the most significant and repeated terms by the interviewees to be counted [39]. This should indicate whether there are coincidences or common patterns in the application of strategies to combat disinformation or verification tools.

The responses of the participants have been coded in this way for open questions, while for closed or dichotomous questions, it was carried out numerically. (See Table 1)

**Table 1.** Questionnaire completed by selected sample during the 38th Annual Conference of CIRCOM held in Galway on 26th an 27 h.

| **Questionnaire** | |
| --- | --- |
| | 1. What are your channel's main strategies in the fight against misinformation? |
| | 2. What fact checking systems or strategies does your channel use?/Are the verification tools you use your own or do they belong to third parties? |
| | 3. What resources have been used to build mechanisms/platforms for verification or detection of fake news? Is there evidence of the use of agents/companies/external collaboration in the development of these tools? Were the verification tools made in house, or do they come from third parties? |
| **Disinformation and verification** | 4. Is there a specific department or area dedicated to verification? #Yes #No |
| | 5. If yes, how many people work in that department? |
| | 6. What is the work process like in the area of disinformation? #Human-based #Software and digital resource-based #Both |
| | 7. What resources are employed to detect fake news? |
| | 8. Misinformation is detected most within which subject matter? #Politics #Health #Economy #Society #Others |
| | 9. Fake news tends to appear within local, regional or national news? #Local #Regional #National |

Source: Prepared by the authors.

The sample selected is made up of the active members of the CIRCOM network, which includes 43 broadcasters from a total of 29 countries. The questionnaire on verification and strategies to combat disinformation was applied to directors of 18 radio and television stations in 15 European countries.

The resulting sample is made up of the following:

ORF (Austria), RTBF (Belgium), HRT (Croatia), YLE (Finland), FTV and France3 (France), HR y RBB (Germany), RAI (Italy), MTVA (Hungary), RTV Oost (The Netherlands), NRK (Norway), RTV (Serbia), RTVS (Slovakia), RTVSLO (Slovenia), EITB and RTVG (Spain) and SVT (Sweden). (See Figure 1).

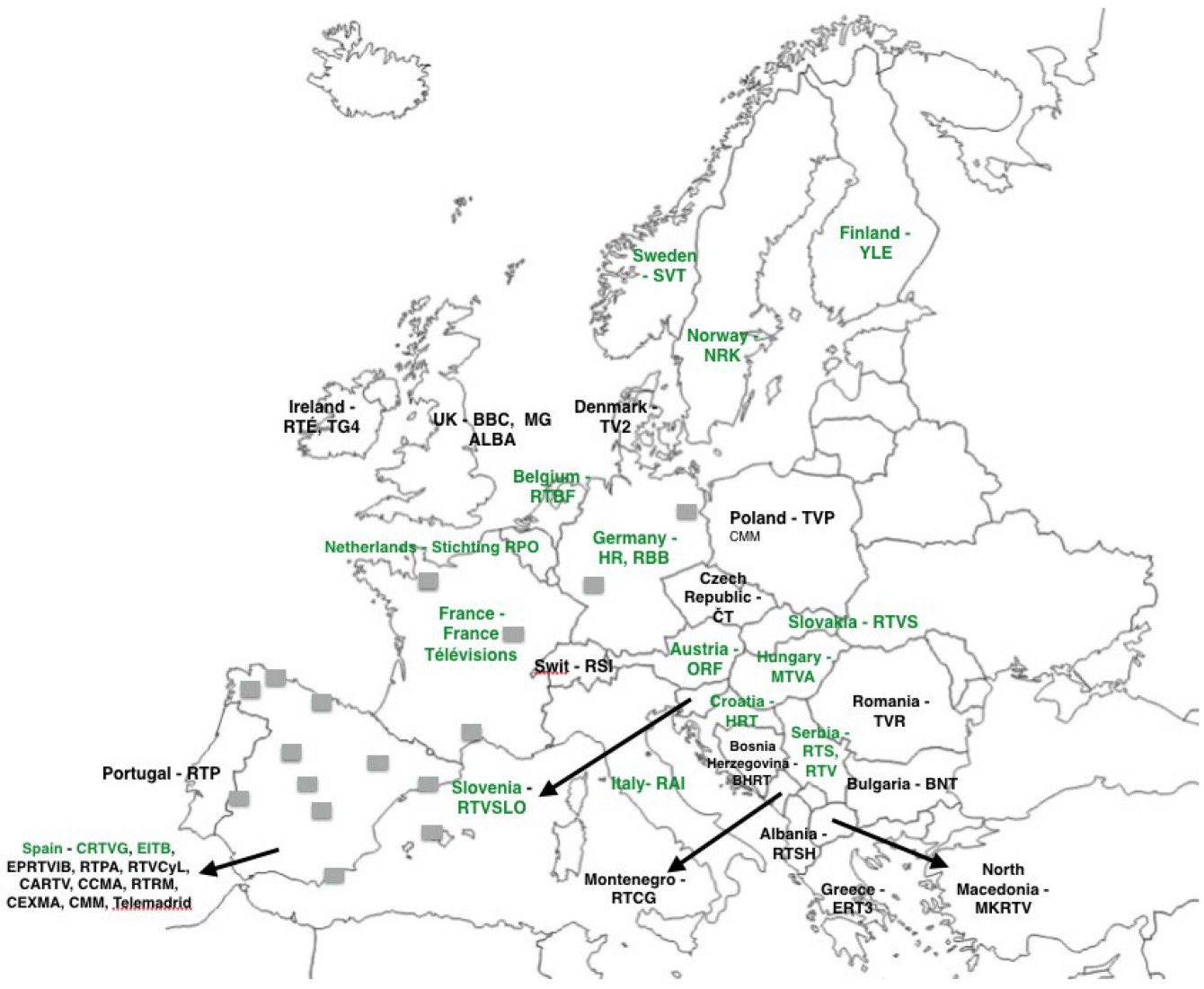

**Figure 1.** Members of CIRCOM network. In green: broadcasters that responded to the questionnaire.

## 5. Results

According to the data collected and the responses of the majority of those interviewed, disinformation is a global problem that also reverberates at local and regional levels. This question is also observed in the literature review and it takes the same approach. Media, academia and social agents are constantly concerned about the diverse consequences that arise from the spreading of false information and how its leak affects the different layers from an international to a local level.

In recent years, terms such as disinformation or fake news have been part of the daily life of the media of any kind. The task of checking and contrasting sources used to be an essential part of the day to day news work. That which seems to be linked to the DNA of the journalist of yesteryear, today becomes a daunting task, given the tons of information that need to be analysed every single day. It is no longer just about comparing sources, it is also necessary to acquire technological skills to be able to implement the advances in the

detection of fake images or videos that are increasingly spread. The issue of disinformation is becoming more sophisticated and requires more resources and knowledge. Although not all the false information broadcasted in the different channels and formats has a harmful origin, the injurious capacity of this practice has been more than verified all over multiple recent political and social events.

When fake news is detected at the local level, many times the source is national or international information. When this information has its origin in a provincial or municipal level and affects in this environment, it has a shorter path since it is usually easier to trace sources and stop transmission, as well as control damage in the population.

Regional media's fight against disinformation constitutes one of the strategic lines included in their decalogues. In the case of the members of the CIRCOM network, given their status as public media and their proximity, the problem of disinformation becomes more relevant due to the close relationship that these broadcasters have with their community. For this reason, the completed questionnaires highlight the relationship and collaboration with the audience as a key factor. There are also other recurring issues in response to the first key question in the research (which deals with strategies to combat disinformation). These include media literacy as a basis for preventing *fake news* from spreading among the population, collaboration with verification initiatives, and the task of contrasting sources, something implicit in the day-to-day of a news channel. (See Table 2)

**Table 2.** The most relevant ideas and themes in relation to the questionnaire.

| 1. What are Your Channel's Main Strategies in the Fight Against Misinformation? | |
| --- | --- |
| **Most relevant and mentioned ideas and terms** | Audience collaboration in detection |
| | Media literacy |
| | Contrasting sources |
| | Collaboration with third parties |

Source: Prepared by the authors.

The concept of media literacy has different approaches among the interviewees. Some media are more willing to consider the training of the population as a key knowledge aimed at increasing their ability to detect false information. Regarding this consideration, itineraries of steps to follow when information is received or tool packages are made available to the user so that any individual can perform a check. Another one of the approaches has a broader meaning and advocates media literacy as viewer training in the current media context.

In many cases, the importance attached to the relationship with the audience is highlighted. Since the reception of false information is passive and through different media or broadcasters, fact-checking requires commitment from the user when somebody wants to verify certain data. Today, the strategies, contents and policies developed by the broadcasters are considerably influenced by their relationship with their audience since this collaboration is essential in this process. It is the viewers themselves who help to detect hoaxes that are disseminated at a certain time. The work of the media not only consists in denying it, but also in replicating the information verified to stop the hoax.

Verification and source contrast as a main strategy to follow arises in almost all the answers when the questions allude to the basic strategies to combat disinformation. It is a concept that is inherently linked to journalistic work and is maintained in this way in newsrooms. The difficulty appears when the volume of sources to be compared is much higher than what can be covered by a single professional and the reliability is harder to determine due to echo chambers. Those changes in the paradigm of disseminated information are what make fake news a bigger problem than it used to be. (See Table 3)

**Table 3.** The most relevant ideas and themes in relation to the questionnaire.

| **2. What Fact-Checking Systems or Strategies Does Your Channel Use?/Are The Verification Tools You Use Your Own or Do They Belong to Third Parties?** | |
| --- | --- |
| **Most relevant and mentioned ideas and terms** | Journalism work approach |
| | Collaboration with EDMO and IFCN initiatives |
| | Spectators training to identify reliable sources |
| | Tv contents and camping to fight against disinformation |

Source: Prepared by the authors.

Regarding verification tasks, 10 of the 18 respondents agree that it mostly comes down to the intrinsic journalistic work of newsrooms. Some state that in addition to systematic checking of sources, collaboration with third parties is crucial, such as institutions, universities or verification initiatives. This includes those that are part of the International Fact-Checking Network (IFCN) [40] or the European Digital Media Observatory (EDMO) [41]. In addition, and correlating with the previous answers, the importance of media literacy is noted as a way to encourage viewers to correctly identify the information. This training is often offered by the channel itself as part of its strategy to alleviate the effects of misinformation on the population.

The majority of radio and television stations surveyed (14 out of 18) stated that they do not have a specific department to tackle disinformation and verification. Clarifying this question further with the interviewees, two fundamental reasons are provided in most cases: lack of resources or dependence on national stations.

In cases where there is a department dedicated to verification, the answers are variable. At France3, this department is shared with other channel headquarters. At NRK, the Norwegian television station, it is a department of about 40 professionals; however, it is a resource that is co-funded with other media outlets and in collaboration with independent verification initiatives. Some of these professionals have IT profiles; therefore, it is a multidisciplinary department. In the case of RAI, the department to combat disinformation has been expanded since the pandemic, made up of between eight and ten professionals. The Belgian television station RTBF has a professional dedicated to verification who belongs to the FAKY initiative, a platform where information can be verified automatically.

The existence or lack thereof of specific departments for disinformation depends on an economic question and, therefore, on the possibility of allocating technological and human resources permanently to this purpose. It also depends on the philosophy that every country has about the fight against disinformation and the importance or interference it has in its own daily life.

Regarding approaches to verification, the questionnaire posed a closed question with three possible options: What is the work process like in the area of disinformation? Human-based/Software and digital resource-based/Both. A total of 10 of the 18 participants agree that, to this day, responsibility lies on the HR since the curation of the information that must be verified involves the selection of the journalist. Those who answered "Both" (6 out of 18) did so in reference to the dependence on the different technological tools that are necessary in the work process. The media and verification agencies concur with the idea that professionals and digital resources have the same importance in order to contrast and deny false information. However, the identification of what disinformation is imperative to verify and what is not is still a human task.

Concerning the collaboration with external agents, the majority of those surveyed state that verification is assumed to be an internal issue, as well as being a task intrinsic to the act of news writing. Despite the fact that the occasionally high volume of disinformation circulation makes verification very difficult, these media outlets' lack of resources makes it impossible to promote their own departments or to allocate personnel for it.

However, the broadcasters that responded to this question differently, indicating that they collaborate with third parties, provide very interesting examples. The Swedish television station SVT works together with International Fact-Checking Network (IFCN) initiatives and with a verifier that belongs to several media outlets. Media, universities and institutions teach how to protect data and personal information and how to locate reliable sources. The Norwegian NRK has developed FAKTISK, its own verifier for the media and individuals, which allows a semi-automatic search and comparison of information. The Slovenian RTVSLO has created its own tool and focuses its strategy on prevention and literacy. The previously mentioned FAKY, a verification platform of the Belgian television RTBF, is an initiative that thrives on collaboration with other institutions such as EDMO, the European Commission or Truly Media, a web-based collaboration platform developed to mainly support journalists and human rights workers in the verification of digital content and material hosted on social networks [42].

Europe is the continent in which the largest number of verification initiatives have been developed. These initiatives act as independent entities in the task of fact-checking and verification. However, their work has become essential for the media, allowing collaboration in fact-checking tasks without having to have large resources or their own departments. Most of these initiatives are grouped in the International Fact-checking Network (INFC). The profiles of these entities are diverse, since they are sometimes financed by governments, or altruistically by users and subventions. Otherwise, they can be funding by the media themselves in organized alliances to provide these services and pay for an entity by covering the budget among various media outlets. In this case, every partner has access to the issuance of the information that the organization verifies, and the hoaxes that are tracked. There are also mixed schemes in which this type of initiatives that are launched as non-profit associations or media companies offer their services in a timely manner for the tracking and verification of some specific information or hoax.

Usually, these third parties collaborations play out similarly to television contents or specific sections on disinformation on news programs or other types of content on current affairs, politics or society. The interviewees agree that the effect of the pandemic on the media in general and on the regional media, in particular, notably increased the need to give space to disinformation and verification. Given the uncertainty caused by the health emergency situation, the role of the local media has been essential in the spread of truthful information and the link with the citizens, showing the reality of each region and the incidence of COVID-19 in each community.

Over the last year (2021–2022), 11 of the 18 respondents state that they have detected false information circulating and that some of this fake news has reached their newsrooms. When asked about the most recurring issues, above all, misinformation is most commonly detected in political content, though there was a marked increase in misinformative health-related content as a result of the COVID-19 pandemic.

The health crisis led to a flood of false information in circulation which, in addition to the fundamental problems in terms of health, was a cross-cutting issue that interfered in politics, society and the economy; for this reason, it ranks as one of the leading topics in the vast majority of surveys.

About the extent to which more misinformation is detected at local, regional or national levels, 9 out of 18 consulted responded that the national level suffers the most, 4 responded that it is the local level where misinformation is most prevalent and 2 reported it in the regional context. The rest of the respondents did not identify one area that evidenced more misinformation than another. Elaborating, they comment on how false information often first occurs at a local level before growing and having a national reach. The opposite can also occur, when misinformation emanates first from national or international spheres and is filtered to regional or local strata, directly affecting those communities.

Given that one of the issues that is identified as a priority in the detection of false information is politics, whether the scope is local or national is relevant depending on what issue or who the target is of the disinformation. In this case, when alluding to issues that only affect a certain region, it is difficult for disinformation to escalate; however, if the attack, hoax, or partially true information may be of general interest and may have a greater scope, it could be filtered to higher levels and circulate at the national or international level. For this reason, as regards this question, as manifested through the survey carried out, it is difficult to identify a unitary answer. According to interviewees, disinformation travels from local to national levels and vice versa. It seems not to be clear which is the level where more disinformation is detected because of the dynamic traceability of misinformation. The conclusion is that more than the field itself, scalability lies in the disinformation in circulation itself and its interest in being shared or not depending on the harmful capacity.

To summarize, it should be said that most of the interviewees emphasize the transversality of the regional media in the field of verification, since, although the lack of resources and dependence on the national delegations is highlighted once again, the relationship established with the audience is closer and more collaborative, especially in the field of information.

## 6. Discussion

Due to massive dissemination through social networks, misinformation is an issue for all media and constitutes a serious problem. In addition, due to limitations related to budget and influence, regional media outlets have fewer resources to fight against fake news. According to Bennet and Livingston [43], in recent years, false information has adopted very sophisticated formulas that significantly increase the journalists' work in the newsroom because the amount of information to check is higher and higher as time goes by. This, in combination with the lack of capability to deal with fake news, is a major problem that undertakes both traditional media and digital native media. The inherence of disinformation and misinformation related to politics and damage on democracies [44] has led to the development of fact-checking initiatives and observatories all around the globe. In Europe, for example, as is mentioned, through EDMO, the IFCN or the efforts of UE to fight the spread on social media and media outlets [45].

Nevertheless, the relationship established between the regional media and their audience is crucial regarding how close this link is. The role of spectators turns out to be of high relevance due to their collaboration in order to identify disinformation that is being disseminated in a certain moment. Once disinformation is located, the first main task is to determine the level of viralisation to start the verification process. Therefore, this dialogue leads to the fact that audience and media are in the same page as regards developing a fundamental collaboration in the fight against misinformation. That is why the regional media consider media literacy and the training of population in this sense an imperative. Although the audience does not have the resources to verify some information, maybe because of a lack of technological skills, maybe because a lack of ability or time to do it, their labour is very important to alert new outlets about information that is being spread and to cut off the chain of misinformation.

## 7. Conclusions

Verification has become an essential part of the development of day-to-day work in newsrooms. The media have a responsibility not to produce, receive or share misinformation. However, given the volume of false information circulating across multiple platforms and its massive consumption and replication by the public, verification has become a strategic issue requiring specific resources. Contrary to popular belief, although there are many tools and technological solutions needed to counteract disinformation, the curation of content and verification of sources continue to be tasks that, for the most part, rely on human efforts.

While not the case for most radio or television stations, those media outlets with the largest budgets or with the highest incidence of false information distributed in their communities have chosen to allocate specific departments for verification, having confirmed that these are necessary measures as the phenomenon of misinformation seems to have a great staying power. In light of recent events in the spheres of international politics and health regarding the pandemic, it is beyond doubt that fake news generates insecurity and damages social well-being.

The framework devised by European regional media through their responses suggests that, as far as verification is concerned, collaboration with third parties is desirable. According to the examples given, the platforms that bring together various entities have shown excellent results in fostering dialogue between the media, the audience and the institutions. In this way, resources from various sources are brought together and joint work is carried out that benefits all those involved. This formula is more effective than each individual newsroom undertaking verification tasks in a unitary way.

In conclusion, those responsible for the broadcasters that are part of the European Association of Regional Television, the CIRCOM network, have shown their growing concern about the need to establish joint strategies to combat disinformation, to comply with the principles of objectivity and veracity that they must uphold. In this way, they are on track to improve the quality of the content generated.

## 8. Limitations and Future Research Directions

The research presented in this article has encountered a series of limitations that condition the results presented here. To begin with, although the sample is significant and of high quality, it is necessary to broaden the results to include representatives of all European regional broadcasters. Although the results could be extrapolated, it would also be interesting to add countries outside the European Union and to segment the results according to the models of Hallin and Mancini [46], since previous research by the authors [47] has found evidence that these models affect the treatment of disinformation. Likewise, during the coding of the information and through the different revisions of the text, shortcomings have been detected in the formulation of the questionnaires that will be solved in future research by adding quantitative variables and value scales in the formulation of the interviews.

**Author Contributions:** Methodology, T.R.-M.; Investigation, J.R.-A. and T.R.-M.; Resources, T.R.-M.; Data curation, J.F.-P.; Writing—original draft, T.R.-M.; Writing—review & editing, J.F.-P.; Supervision, J.R.-A. All authors have read and agreed to the published version of the manuscript.

**Funding:** This article is part of the activities of the projects: "FAKELOCAL: Map of Disinformation in the Autonomous Communities and Local Entities of Spain and its Digital Ecosystem" (Ref. PID2021-124293OB-I00), funding by the Ministry of Science and Innovation, the State Research Agency (AEI) of the Government of Spain and by the FEDER of the European Union. "VALCOMM: Public audiovisual media before the platform ecosystem: management models and evaluation of the reference public value for Spain" (PID2021-122386OB-I00), funding by MCIN, AEI and FEDER, UE.

**Institutional Review Board Statement:** Not applicable.

**Informed Consent Statement:** Not applicable.

**Data Availability Statement:** Not applicable.

**Conflicts of Interest:** The authors declare no conflict of interest.

## Notes

[1]   The acronym "CIRCOM" represents a declaration of intent, worded in French and abbreviated like this: "Cooperative Internationale de Recherche et d'Action en matière de Communication (International Cooperative for Research and Action on the Field of Communication). That long description was abbreviated for practical reasons to six capital letters: CIRCOM. (Source: https://www.circom-regional.eu/about-circom-3/about-circom-l) (accessed on 16 January 2023).

[2]   The acronym "CIRCOM" represents a declaration of intent, worded in French and abbreviated like this: "Cooperative Internationale de Recherche et d'Action en matière de Communication (International Cooperative for Research and Action on the Field of Communication). That long description was abbreviated for practical reasons to six capital letters: CIRCOM. (Fuente: https://www.circom-regional.eu/about-circom-3/about-circom-l) (accessed on 16 January 2023).

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
