# Peer review of "Disinformation and Verification in a Digital Society: An Analysis of Strategies and Policies Applied in the European Regional TV Broadcasters of the CIRCOM Network"

_societies, doi:10.3390/soc13040081_

Round 1

Reviewer 1 Report

A description of PSM (chapter 2) would be useful to the readers.

Author Response

A definition od PSM has been included

Reviewer 2 Report

We congratulate the authors for the proposal presented, which is an important contribution insofar as there is a scarcity of studies on disinformation at regional or local levels.

Two brief contributions, for improvement:

1. There are newer versions of Digital News Report;

2. Due to the scarcity of studies and possible lack of knowledge of their existence, we leave the following works to the authors' consideration:

- Park, J. (2021). Survival strategies: How journalism is innovating to find sustainable ways to serve local communities around the world and fight against misinformation. International Press Institute. Available online: https://ipi.media/local-journalism-report-survival-strategies

- Jeronimo, P.; Esparza, MS. (2022). Disinformation at a Local Level: An Emerging Discussion. Publications, 10, 15. https://doi.org/10.3390/publications10020015

Reviewer 3 Report

The theme is interesting, relevant and current, and still little explored from a scientific point of view.

It makes reference to the issue of disinformation particularly in the context of the pandemic, but I think it could be interesting to also include the theme of the Russian-Ukrainian war in this framework.

The introduction should be better developed, exposing the topic under analysis in a more concrete way, as well as the object of study.

The theoretical foundation deserves to be better developed. The problem of disinformation is not exclusive to the pandemic period nor does it circulate only in the conventional media. It would be interesting to contextualize the issue of disinformation more concretely in the context of the Internet and social media. This issue is referenced, but not properly detailed. The very conceptual approach to the work seems to me insufficient and too superficial. What are the concepts and theoretical dimensions of analysis that frame this investigation?

Overall, the theoretical foundation needs considerable improvement. The text that is presented is a good starting point, however, it also presents a technical view of the problem of disinformation, addressing little or nothing the impact of disinformation in social contexts. A more humanizing view of the object of study would enrich the presented text.

With regard to methodology, the study has a good sample size, however it would be interesting to detail a little more about the instrument's design, namely which concepts, dimensions and indicators are at the basis of its design. At the same time, the research questions are unclear, as well as their concrete objectives. I think that this fragility in the research design dimension may be at the origin of the difficulty in presenting, strictly speaking, the process of conceptualizing the research instrument.

The presentation of the results presents interesting data, however there is no real dialogue with the presented literature. The conclusions are also not robust enough, not existing to point out, also, the dimensions of analysis of future investigation that this work starts.

Overall, the work presented is still in an exploratory phase that deserves, however, to be better developed.

Author Response

Point 1: The theme is interesting, relevant and current, and still little explored from a scientific point of view.

It makes reference to the issue of disinformation particularly in the context of the pandemic, but I think it could be interesting to also include the theme of the Russian-Ukrainian war in this framework.

The introduction should be better developed, exposing the topic under analysis in a more concrete way, as well as the object of study.

Response 1: The authors has tried to introduce some references to enhance the article in this sense and be clear to the readers that the approach is exploratory and the object of study is the CIRCOM network. We already added a reference to the Russian-Ukrainian war. 

Point 2: The theoretical foundation deserves to be better developed. The problem of disinformation is not exclusive to the pandemic period nor does it circulate only in the conventional media. It would be interesting to contextualize the issue of disinformation more concretely in the context of the Internet and social media. This issue is referenced, but not properly detailed. The very conceptual approach to the work seems to me insufficient and too superficial. What are the concepts and theoretical dimensions of analysis that frame this investigation?

Response 2: The authors perfectly understand this comment but the Circom Network and the media outlets involved are more related to linear tv. Given that, the study is focused is the tradicional media and their approach to innovation and the management of disinformation in the digital enviroment.

Point 3: Overall, the theoretical foundation needs considerable improvement. The text that is presented is a good starting point, however, it also presents a technical view of the problem of disinformation, addressing little or nothing the impact of disinformation in social contexts. A more humanizing view of the object of study would enrich the presented text.

Response 3: We would take this into consideration for further questionaries.

Point 4: With regard to methodology, the study has a good sample size, however it would be interesting to detail a little more about the instrument's design, namely which concepts, dimensions and indicators are at the basis of its design. At the same time, the research questions are unclear, as well as their concrete objectives. I think that this fragility in the research design dimension may be at the origin of the difficulty in presenting, strictly speaking, the process of conceptualizing the research instrument.

Response 4: this comment helps us a lot for the development of the next questionaries. We´ll try to improve the research questions as well the objectives of the interviews.

Reviewer 4 Report

There is an error in the title of item 4 (Methology) --> Methodology

The article does not formulate research hypotheses. It focuses on an exploratory approach. This should have been made clearly in the introduction.

The questionnaire lacks a question that explicitly refers to the use of AI and self-learning algorithms in the verification of fake news. This theme is hinted at in question 6, but too generally. Meanwhile, in the article, the authors discuss only the first 2 questions.

The article as a whole is correct, with valid conclusions.

Round 2

Reviewer 3 Report

I do not believe that the authors have considered the comments made.

The work does not present a substantial revision to what was requested.

Author Response

(The authors gave the same response as above.)
